# RADIAL SPIKE AND SLAB BAYESIAN NEURAL NETWORKS FOR SPARSE DATA IN RANSOMWARE ATTACKS

Ransomware attacks are increasing at an alarming rate, leading to large financial losses, unrecoverable encrypted data, data leakage, and privacy concerns. The prompt detection of ransomware attacks is required to minimize further damage, particularly during the encryption stage. However, the frequency and structure of the observed ransomware attack data makes this task difficult to accomplish in practice. The data corresponding to ransomware attacks represents temporal, high-dimensional sparse signals, with limited records and very imbalanced classes. While traditional deep learning models have been able to achieve state-of-the-art results in a wide variety of domains, Bayesian Neural Networks, which are a class of probabilistic models, are better suited to the issues of the ransomware data. These models combine ideas from Bayesian statistics with the rich expressive power of neural networks. In this paper, we propose the Radial Spike and Slab Bayesian Neural Network, which is a new type of Bayesian Neural network that includes a new form of the approximate posterior distribution. The model scales well to large architectures and recovers the sparse structure of target functions. We provide a theoretical justification for using this type of distribution, as well as a computationally efficient method to perform variational inference. We demonstrate the performance of our model on a real dataset of ransomware attacks and show improvement over a large number of baselines, including state-of-the-art models such as Neural ODEs (ordinary differential equations). In addition, we propose to represent low-level events as MITRE ATT&CK tactics, techniques, and procedures (TTPs) which allows the model to better generalize to unseen ransomware attacks.

## 1 INTRODUCTION

Ransomware attacks are increasing rapidly and causing significant losses to governments, corporations, non-governmental organizations, and individuals. The losses may include financial costs due to ransoms paid to decrypt assets, unrecoverable files when the ransom is not paid or the attacker fails to provide the decryption key, privacy and intellectual property theft when assets are exported, and even significant injury when ransomware impairs health care devices or patient records in hospitals. It is clear that the timely detection of ransomware incidents is necessary in order to minimize the number of assets that are encrypted or exfiltrated (Urooj et al., 2021). To improve the ransomware response, this work proposes a new Bayesian Neural Network model that offers improved detection rates for organizations which employ analysts to protect their assets and networks.

The problem is usually considered as a detection task, where the two classes are ransomware or not. The traditional methods of statistics and machine learning have been proposed to detect security threats in general and specifically ransomware in some cases. From the statistical perspective, a common approach is the application of Bayesian Networks (Perusquía et al., 2020; Oyen et al., 2016; Shin et al., 2015), whose main goal is to model the relationship between the observed signal and the class of the attack as a graphical model. From the machine learning perspective, a range of models were used to detect ransomware (Alhawi et al., 2018; Poudyal et al., 2018; Zhang et al., 2019; Larsen et al., 2021), such as Naive Bayes, Gradient Boosting, and Random Forests.

**Bottleneck.** To obtain the rich expressive power of traditional deep learning models, training usually requires having access to a large number of records to successfully obtain robust generalized results. Unfortunately, the frequency and structure of commonly observed data corresponding to ransomware attacks makes this task more difficult to accomplish. In particular, ransomware attack data can be represented as temporal high-dimensional sparse signals, with a limited number of records and very imbalanced classes. In our data, the percentage of ransomware attacks to non-ransomware attacks is 1% versus 99%, respectively.

**Main ideas and contributions.** To address these unique features of the ransomware data, we first propose to represent ransomware signals according their MITRE ATT&CK tactics, techniques, and procedures (TTPs) which allows us to generalize ransomware and other attacks at a higher-level instead of the low-level detections associated with an individual attack. In addition, this allows for the detection of both human operated and automated ransomware attacks across multiple stages in the kill chain within an organization's network. Next, we propose a new probabilistic model which is called the Radial Spike and Slab Bayesian Neural Network. It is a Bayesian Neural Network, where the approximate posterior is represented by a mixture of distributions, resulting in a Radial Spike and Slab distribution. Our model provides the following benefits including: (1) the Spike and Slab component handles missing or sparse data, (2) the Radial component scales well with the growth of the number of parameters in the deep neural network, and (3) the Bayesian component prevents overfitting in the limited data setup. From the theoretical perspective, we provide the justification for using this type of distribution, as well as a computationally efficient method to perform variational inference. In the results section, we demonstrate the performance of our model on a set of actual ransomware attacks and show improvement over a number of baselines, including the state-of-the-art temporal models such as RNNs (Cho et al., 2014) and Neural ODEs (ordinary differential equations) (Chen et al., 2018). Thus, the proposed model is an important tool for the critical problem of ransomware detection.

## 2 INCIDENT DATA DESCRIPTION

This work utilizes threat data provided by 'our industry partner' to detect ransomware and other types of cybersecurity attacks. Low-level event generators are manually created by analysts (i.e., signatures) and are provided with a UUID (Universally Unique Identifier).

**Features.** Given each incident, features need to be extracted which capture the range of attack behaviors observed across the kill chain and represent common behaviors across the different families of ransomware attacks. The low-level events cannot be used directly because there are too many to train our model, given the number of labeled examples, and they do not generalize well individually. To overcome these problems, we map a subset of the low-level events into a higher-level representation using the MITRE ATT&CK framework (MITRE). We chose the MITRE ATT&CK framework for the mapping because it provides a knowledge base of adversary tactics, techniques, and procedures (TTPs) and is widely used across the industry for classifying attack behaviors and understanding the lifecycle of an attack. Using the MITRE ATT&CK TTPs is a natural choice for features as it is generalizable, interpretable, and easy to acquire for this data as each low-level event from 'the anonymized company' is tagged with the MITRE technique associated with the alerted behavior (MITRE). For example, one of the features can represent whether 'OS Credential Dumping' happened or not. Additional MITRE ATT&CK features are included in Table 2, and the entire set is provided by the MITRE corporation (MITRE, 2022a). The verbose definition of these features can be found in (MITRE). For example, feature T1059.001 "Command and Scripting Interpreter, Powershell" corresponds to "Adversaries may abuse PowerShell commands and scripts for execution" (MITRE, 2022b). In total, our data is a sparse binary, high-dimensional vector of size 706, which contains 298 MITRE ATT&CK features and 408 additional signature-based features, at each time point. One of the primary characteristics of the data is sparsity because only very few actions are completed at each time step during the attack.

**Labels.** Using manual investigation, an analyst provides a label for each incident indicating whether it is due to a ransomware attack or another type of attack. The ransomware incidents include both human operated ransomware (HumOR) and automated ransomware attacks described in Appendix B in the Supplementary Material. However, our positive class label only indicates that an attack is ransomware and does not distinguish between the two classes of ransomware (i.e., HumOR, Automated). Our goal is to build an alarm-recommendation system, which can not only detect a possible ransomware attack, but also provide an estimate of the uncertainty about the decision. We provide additional details about the training and testing data in Section 4.

**Ethics.** As part of the production data collection process, all data has been processed to remove all personal identifiable information. The datasets we received for this analysis only included a randomly assigned UUID for the organization, and the incidents that included the MITRE events,

signature UUID identifiers and the labels. The data was collected and adheres to the GDPR standard. There are no negative societal impacts for creating models to protect users from ransomware.

## 3 METHODOLOGY

Important features of probabilistic models, such as providing a notion of uncertainty, dealing with missing data, and preventing overfitting in a limited data regime, have generated a strong interest in deep Bayesian learning. In this section, we provide more details regarding Bayesian Neural Networks, including different aspects of initializing and training the model. We then propose the Radial Spike and Slab Bayesian Neural Network model to address the problems of the ransomware data.

**Bayesian Neural Networks.** The main idea behind the Bayesian Neural Network is to consider all weights as being samples from a random distribution. Formally, we denote the observed data as $(x, y)$, where $x$ is an input to the network, and $y$ is a corresponding response. Let all weights of a BNN, $W = (W^1, \ldots, W^D)$, be a random vector, where $D$ is the depth (i.e., number of layers) of the BNN and each $W^j = (w^{j,1}, \ldots, w^{j,l_j})$ is a random vector itself of all weights $w^{j,k}$ per layer $W^j$ of size $l_j$. To generate uncertainty of the prediction, we need to be able to compute $p(y|x)$. However, since all weights of a BNN are considered to be random variables, we can rewrite the conditional probability as $p(y|x) = \int_w p(y, W|x)dW = \int_W p(y|W, x)p(W|x)dW$. Typically, the likelihood term $p(y|W, x)$ is defined by the problem setup, e.g., if we consider classification, as in ransomware incident detection, $y \sim Bern(g(W, x))$ for some function $g$. Then, the main problem of training a BNN is to compute the posterior probability $p(W|x)$, given the observed data $x$ and a suitable prior for $W$.

In some simple cases of small neural networks, it may be possible to obtain a closed-form solution for the posterior if the prior and posterior are conjugate distributions. In other cases, if a closed-form solution is unavailable, sampling-based strategies are required such as Markov Chain Monte Carlo schemes based on Gibbs or Metropolis Hasting samplers. While such an approach provides excellent statistical behavior with theoretical support, scalability as a function of the dimensionality of the problem is known to be a serious issue. The alternative for machine learning and vision problems is Variational Inference (VI) (Graves, 2011). The core concept of VI is based on the fact that approximating the true posterior with another distribution may often be acceptable in practice. The computational advantages of VI permit estimation procedures in cases which would not otherwise be feasible. VI is now a mature technology, and its success has led to a number of follow-up developments focused on theoretical as well as practical aspects (Blundell et al., 2015).

When using VI in Bayesian Neural Networks, we approximate the true unknown posterior distribution $P(W|x)$ with an *approximate posterior* distribution $Q_\theta$ of *our choice*, which depends on learned parameters $\theta$. Let $W_\theta = (W_\theta^1, \ldots, W_\theta^D)$ denote a random vector with distribution $Q_\theta$ and probability distribution function (pdf) $q_\theta$. VI seeks to find $\theta$ such that $Q_\theta$ is as close as possible to the real (unknown) posterior $P(W|x)$, and this is accomplished by minimizing the Kullback–Leibler ($KL$) divergence between $Q_\theta$ and $P(W|x)$. Given a prior pdf of weights, $p$, with a likelihood term $p(y|W, x)$, and the common *mean field* assumption of independence for $W^d$ and $W_\theta^d$ for $d \in 1, \ldots, D$, i.e., $p(W) = \prod_{d=1}^{D} p^d(W^d)$ and $q_\theta(W_\theta) = \prod_{d=1}^{D} q_\theta^d(W_\theta^d)$,

$$\boldsymbol{\theta}^* = \arg\min_{\theta} KL(q_\theta||p) - \mathbb{E}_{q_\theta}[\ln p(y|W, x)] \tag{1}$$

$$KL(q_\theta||p) = \sum_{d=1}^{D} \mathbb{E}_{q_\theta^d}[\ln q_\theta^d(w)] - \mathbb{E}_{q_\theta^d}[\ln p^d(w)]. \tag{2}$$

By definition of the expected value $\mathbb{E}_{q_\theta}$, it is necessary to compute the multi-dimensional integral w.r.t $w \sim Q_\theta$ to solve equation 1. If such integrals are impossible to compute in a closed-form, a numerical approximation is used (Ranganath et al., 2014; Paisley et al., 2012; Miller et al., 2017). For example, Monte Carlo (MC) sampling yields an asymptotically exact, unbiased estimator with

variance $O(\frac{1}{M})$, where $M$ is the number of samples. For a function $g(\cdot)$:

$$\mathbb{E}_{q_\theta}\left[g(w)\right] = \int g(w)q_\theta(w)dw \approx \frac{1}{M}\sum_{i=1}^{M}g(w_i), \text{ where } w_i \sim Q_\theta. \qquad (3)$$

The expected value terms in equation 1 and equation 2 can be estimated by applying the method in equation 3, and in fact, even if a closed-form expression can be computed, an MC approximation may perform similarly given enough samples (Blundell et al., 2015).

Given a mechanism to solve equation 1, the main consideration in VI is the *choice of prior $p$ and the approximate posterior $q_\theta$*. A common choice for $p$ and $q_\theta$ is Gaussian, which allows calculating equation 2 in a closed-form. However, this type of distribution is mainly used for computational purposes and does not reflect the nature of the data. Choosing the correct distribution, especially the one which can incorporate the features of the analyzed data, is an open problem (Ghosh & Doshi-Velez, 2017; Farquhar et al., 2020; McGregor et al., 2019; Krishnan et al., 2019). In the next section, we discuss our proposed distribution, which naturally fits the data encountered in ransomware incident detection. While we give the description of the analyzed data in Section 2, we next describe the features of the data, which are important to encapsulate in the model design.

**Spike and Slab distribution.** The sparsity of the data is a common problem in many areas (Kang, 2013) and was previously approached from different perspectives. For example in the statistics community, sparsity can be addressed with both Stochastic Regression Imputation and Likelihood Based Approaches (Lakshminarayan et al., 1999). In the machine learning community, methods based on k-nearest neighbor (Batista & Monard, 2003) and iterative techniques (Buuren & Groothuis-Oudshoorn, 2010) have been developed, including approaches with neural networks (Sharpe & Solly, 1995; Śmieja et al., 2018). Another way to tackle sparsity comes from regularization theory via L1 regularization, e.g., group LASSO (Meier et al., 2008), sparse group LASSO (Simon et al., 2013) and graph LASSO (Jacob et al., 2009).

However, we are interested in a probabilistic approach to address the sparsity in our data. From the probabilistic perspective, a common way to account for sparsity of the data in the model is to consider an appropriate distribution. For example, the distribution can be the Horseshoe distribution (Carvalho et al., 2009) or derivatives of the Laplace distribution (Babacan et al., 2009; Bhattacharya et al., 2015). Namely, in our case, we would like to model sparse data with a sparse probabilistic Bayesian neural network. Since only a portion of the input variables are relevant to the response variable, we want the weights to be represented as on/off switches to understand whether we should account for the input variables. Such a sparse Bayesian neural network can be represented by a 'sparse' distribution on its weights, e.g., the mixture of priors with Spike and Slab components which have been widely used for Bayesian variable selection (Mitchell & Beauchamp, 1988; George & McCulloch, 1997). In general, the form of the Spike and Slab distribution for random variable $w$ can be written as: $w \sim (1-\pi)\delta_\xi + \pi g$, where $\pi$ determines the probability for each mixture component, $\delta$ is spike component, which is modeled with a Dirac delta function such that $\delta(w) = \begin{cases} +\infty, & w = \xi \\ 0, & w \neq \xi \end{cases}$ and $\int_{-\infty}^{\infty}\delta(w)dw = 1$, and $g$ is the slab component, which is a general distribution of the practioner's choice. The general idea is to explicitly introduce the sparsity component in the distribution of the data, allowing the probability mass to fully concentrate on $\xi = 0$ with probability $1 - \pi$, and with probability $\pi$ spread the remaining mass over the domain of the slab component $g$. Notice, that $\pi$ can be considered as a random variable itself, e.g., $\pi \sim Bern(\lambda)$, where $\lambda$ is either a learned parameter or a fixed value that is provided by a specialist.

The next questions to consider include: (1) how can the 'Spike and Slab' distribution be applied in a BNN, and (2) which slab component $g$ should we consider?

**Spike and Slab BNN.** In the BNN, all of the neural network's weights $W$ are considered to be random variables, and to use VI to solve equation 1 for each layer's set of weights $W^j$ in $W = (W^1, \ldots, W^D)$, it is necessary to provide the prior $p^j$ and the approximate posterior $q_\theta^j$. Without loss of generality, we consider a single weight $w := w^{j,k}$, dropping the indices $j$ and $k$, and only work with the prior $p$ and the approximate posterior $q$ for the remainder of this section.

Incorporating a Spike and Slab distribution on both the prior $p$ and the approximate posterior $q$, samples $w_p$ from $p$ and $w_q$ from $q$ have the following distribution:

$$w_p|\pi_p \sim (1 - \pi_p)\delta_0 + \pi_p g_p \text{ and } w_q|\pi_q \sim (1 - \pi_q)\delta_0 + \pi_q g_q, \tag{4}$$

where $\pi_p \sim Bern(\lambda_p), \pi_q \sim Bern(\lambda_q)$, and $g_p, g_q$ are distributions of our choice.

As we discussed previously, the main goal of VI is to learn parameters $\theta$ of an approximate posterior $q_\theta$, by minimizing equation 2. In the case of equation 4, $\theta = (\lambda_q, \theta_q)$, where $\lambda_q$ is the probability of the Bernoulli distribution associated with $\pi_q$, and $\theta_q$ are the parameters of the Slab component $g_q$. First, we state Theorem 3.1, which allows us to compute the $KL$ term between two general Spike and Slab distributions.

**Theorem 3.1.** *Given two general Spike and Slab distributions such that: $p(w|\pi_p) = (1-\pi_p)\delta_0(w) + \pi_p g_p(w)$, $q(w|\pi_q) = (1 - \pi_q)\delta_0(w) + \pi_q g_q(w)$, $\pi_p \sim p(\pi) = Bern(\lambda_p)$, and $\pi_q \sim q(\pi) = Bern(\lambda_q)$, with $\delta_0$ being a dirac delta function at 0 and $g_p, g_q$ are the pdfs of the distributions of our choice, the $KL\left(q(w, \pi)\|p(w, \pi)\right)$ is equal to:*

$$KL\left(Bern(\lambda_q)\|Bern(\lambda_p)\right) + \lambda_q KL\left(g_q\|g_p\right). \tag{5}$$

The proof is shown in Appendix F.

**Choice of $g_q$ and $g_p$: Radial distribution.** So far, we have shown results for a general Spike and Slab distribution. The important question is which slab components $g$ should we consider for our approach, and if $g_q$ and $g_p$ should be from the same family? Authors in (Bai et al., 2020) considered both $g_q$ and $g_p$ to be the Gaussian distribution. However, there is emerging evidence (Farquhar et al., 2020; Fortuin et al., 2020) that the Gaussian assumption results in poor performance of the medium to large-scale Bayesian Neural Networks. Authors regard this as being caused by the probability mass in a high-dimensional Gaussian distribution concentrating in a narrow "soap-bubble" far from the mean. For this reason, (Farquhar et al., 2020) proposed using a Radial distribution with parameters $(\mu, \sigma)$, where samples can be generated as:

$$\mu + \sigma * \frac{\xi}{\|\xi\|} * |r| \sim Radial(\mu, \sigma), where \xi \sim MVN(0, I), r \sim N(0, 1). \tag{6}$$

Following (Farquhar et al., 2020), we set up our approximate posterior $g_q$ to be the Radial distribution $(\mu, \sigma)$, while the prior $g_p$ is Normal(0, 1). Given equation 5, it is necessary to define the $KL\left(g_q\|g_p\right)$ term. Unfortunately, a closed-form solution for our choice of $g_q$ and $g_p$ is not available, and we approximate the $KL$ term using Monte Carlo sampling from equation 3 with $M$ samples. This process leads to (up to a constant): $KL\left(g_q\|g_p\right) \approx -\log \sigma - \frac{1}{M}\sum_{i=1}^{M} \log[p(w_i)]$, where $w_i$ is sampled from the Radial distribution $(\mu, \sigma)$ as described in equation 6 and $p$ is the Likelihood of $N(0, 1)$. Note that running an MC approximation for large $M$ can lead to running out of memory in either a GPU or RAM, (Nazarovs et al., 2021). To tackle this issue, we follow (Nazarovs et al., 2021) and apply a graph parameterization for our Radial Spike and Slab distribution, allowing us to set $M = 1000$ without exhausting the memory.

**Reparameterization trick: Gumbel-Softmax.** Given Theorem 3.1, we can rewrite equation 1 as:

$$\boldsymbol{\theta}^* = \underset{\theta=(\lambda_q,\theta_q)}{\arg\min}\ KL\left(Bern(\lambda_q)\|Bern(\lambda_p)\right) + \lambda_q KL\left(g_q\|g_p\right) - \mathbb{E}_{q_\theta}\left[\ln p(y|W, x)\right]. \tag{7}$$

Recall, we can compute the $KL\left(Bern(\lambda_q)\|Bern(\lambda_p)\right)$ in a closed-form (inside the proof of Theorem 3.1) and approximate the $KL\left(g_q\|g_p\right)$ term with MC sampling. Next, there are two main aspects left for our attention: (1) computing $\mathbb{E}_{q_\theta}\left[\ln p(y|W, x)\right]$, which is usually approximated with Monte-Carlo sampling (Kingma & Welling, 2013) because of the intractability issue, and (2) how to do back-propagation for optimization. The problem with back-propagation in this setting is that sampling directly from, e.g., $w \sim N(\mu, \sigma)$ with learnable parameters $\mu$ and $\sigma$, does not allow us to back-propagate through those parameters, and thus, they cannot be learned. This issue is addressed by applying a local-reparameterization trick (Kingma et al., 2015). For example, instead of sampling from $w \sim N(\mu, \sigma)$ directly, we sample $z \sim N(0, 1)$ and compute: $w = \mu + \sigma z$. This allows back-propagation to optimize the loss w.r.t. $\mu$ and $\sigma$.

While the local-reparameterization trick is obvious for members of a location-scale family, like the Gaussian distribution, and even for the selected Radial distribution, it is not clear how to apply this trick to the Bernoulli distribution, $Bern(\lambda)$. One way to address this issue is to approximate samples from the Bernoulli distribution with the Gumbel-Softmax (Maddison et al., 2016; Jang et al., 2016; Bai et al., 2020). That is, $\pi \sim Bern(\lambda)$ is approximated by $\widetilde{\pi} \sim$ Gumbel-Softmax $(\lambda, \tau)$, where $\widetilde{\pi} = (1 + \exp(-\eta/\tau))^{-1}$, $\eta = \log \frac{\lambda}{1-\lambda} + \log \frac{u}{1-u}$, and $u \sim \mathcal{U}(0,1)$. Here, $\tau$ is the parameter which is referred as the temperature. When $\tau$ approaches $0$, $\tilde{\pi}$ converges in distribution to $\pi$. However, in practice, $\tau$ is usually chosen no smaller than $0.5$ for numerical stability (Bai et al., 2020). Applying the Gumbel-Softmax approximation instead of optimizing the loss for parameter $\lambda_q$, we consider a new parameter $\theta_\pi = \log \frac{\lambda_q}{1-\lambda_q}$. Thus, $\lambda_q = S(\theta_\pi) = \frac{1}{1+e^{-\theta_\pi}}$, resulting in the final learned parameters: $\theta = (\theta_\pi, \theta_q)$.

**Final Loss and Method Summary.** A step-by-step summary of the method in provided in Algorithm 1. The final loss is given in Algorithm 2.

---

**Algorithm** 1: Learning the posterior distribution of a BNN $p(W|x)$ with a Radial Spike and Slab approximate posterior, to account for sparsity of the data.

**Input:**
1: Neural Network of depth $D$ with
2: Weights $W_\theta = (W_\theta^1, \ldots, W_\theta^D)$, which have
3: Spike and Slab Radial distribution $Q_\theta$ with pdf $q_\theta$, s.t.
   - $q(w|\pi_q) = (1 - \pi_q)\delta_0(w) + \pi_q g_q(w; \mu, \sigma)$,
   - $g_q(w; \mu, \sigma)$ is pdf of $Radial(\mu, \sigma)$
   - $\pi_q \sim Bern(S(\theta_\pi))$, where $S$ is the softmax, and
4: Prior Spike and Slab distribution $P_\theta$ with pdf $p$, s.t.
   - $p(w|\pi_p) = (1 - \pi_p)\delta_0(w) + \pi_p g_p(w; \mu_p, \sigma_p)$,
   - $g_p(w; \mu_p, \sigma_p)$ is pdf of Gaussian distribution
   - $\pi_p \sim Bern(\pi_p)$

**Output:** Learned parameters $\theta = (\theta_\pi, \mu, \sigma)$

**Require:** Prior distribution's parameters $(\pi_p, \mu_p, \sigma_p)$

5: **while** $\theta$ has not converged **do**
6:     Minimize VI loss in equation 8, by using gradient descent algorithms (e.g., SGD or Adam) and doing:
7:     **Forward pass**: to compute
   - $y$ with local reparameterization trick for both Radial and Bernoulli (using Gumbel-Softmax)
   - $KL$ terms and expected log-likelihood term, using combination of closed-form and MC
8:     **Backward pass**: compute gradients of $\theta$
9: **end while**

---

**Algorithm 2:** Final loss used for optimization in Algorithm 1.

Original:    $KL(Bern(\lambda_q)\|Bern(\lambda_p)) + \lambda_q KL(g_q\|g_p) - \mathbb{E}_{Q_\theta}[\ln p(y|W,x)]$

Final:
$$L = \sum_{\substack{j=1,\ldots,D,\\ k=1,\ldots,l_j}} KL_{j,k} - \mathbb{E}_{Q_\theta}[\ln p(y|W,x)], \text{ where} \tag{8}$$

$KL_{j,k} = (1-S(\theta_\pi^{j,k}))\log\frac{1-S(\theta_\pi^{j,k})}{1-\lambda_p^{j,k}} + S(\theta_\pi^{j,k})\log\frac{S(\theta_\pi^{j,k})}{\lambda_p^{j,k}} + S(\theta_\pi^{j,k})\left\{-\log\sigma^{j,k} - \frac{1}{M}\sum_{i=1}^M \log[p(\mathbf{w_i^{j,k}})]\right\}$

Note that based on the *mean field* assumption of a BNN, the final loss $L$ includes the sum over all $KL_{j,k}$ terms, which are computed for each $k$-th weight $w^{j,k}$ of the $j$-th layer of the BNN with parameters $\theta^{j,k} = (\theta_\pi^{j,k}, \mu^{j,k}, \sigma^{j,k})$. In this case, the final set of trainable parameters is $\theta = \{\theta^{j,k}\}$ for $j = 1, \ldots, D$ and $k = 1, \ldots, l_j$. In addition, $\mathbb{E}_{Q_\theta}$ can be computed either in a closed-form or approximated by MC, depending on the complexity of the BNN.

---

## 4 EXPERIMENTS

**Data description.** As described previously in Section 2, each incident is represented by a temporal sequence of events from a knowledge base of TTPs with an assigned label, which indicates whether it is ransomware or another type of attack. First, the company provided 201 incidents labeled as Ransomware and 24,913 with Non-Ransomware labels for the initial dataset. All of the samples in this dataset were deduplicated and included 706 sparse binary features. This first dataset was randomly split with 80% of the examples assigned to the training set, while the remainder were used to create a validation set. Second, for the test set, we received a newer, deduplicated dataset making it independent of the training and validation sets. This dataset included 644 Ransomware incidents and 14,696 Non-Ransomware incidents.

**Preprocessing of temporal information.** Some of the models such as the Neural ODE benefit from knowledge of the actual time associated with the recorded event, while others, including the RNN with a GRU cell, can be trained on the event sequence based solely on the event index (i.e., t=1,2,...). Finally, other models such as the fully connected and Bayesian Neural Networks can be trained and tested using the aggregation of all of the events in the event sequence. To reduce the number of time steps for the time-based models for our study, we aggregated all TTP events observed within a one minute window. We set the aggregation time to one minute after doing hyperparameter tuning on this value. This results in very few signals being recorded per aggregated time step, which is represented in Figure 1 in Appendix D. We see that the majority of the data have a small number of features that are set, namely less than 10 out of 706 possible. For the neural network models, we aggregated all of the TTP features into a single input vector. All of the sequences for the training and testing datasets were truncated after one hour from the time of the first event.

**Models.** In the experiments, we consider several baseline models from the traditional, temporal, and probabilistic deep learning settings, in addition to our proposed model. From the *temporal perspective*, we consider two models including the Recurrent Neural Network with a GRU cell (RNN) and the Neural ODE (NODE). As we mentioned earlier, the traditional recurrent neural network models (e.g., Simple RNN, GRU, LSTM) ignore the value of the time steps and only consider the order (i.e., index), in contrast to the Neural ODE which accounts for the time step value. *Note*, we originally considered several temporal models, which do not account for the time value, like the traditional (i.e., Simple) RNN, the RNN with a GRU cell, the LSTM, and the Bi-directional LSTM. However, among all of these models, the RNN with the GRU cell performed the best, and we only include this model in the analysis below.

In addition we consider the traditional fully connected neural network (FC), and four BNN models. The first two BNNs are the standard BNNs which have a Gaussian or Radial approximate posterior (BNN: Gaus, BNN: Radial), and the other two are the corresponding Spike and Slab versions, BNN: Spike-Slab Gaussian and our proposed BNN: Spike-Slab Radial. For these networks, we ignore the temporal aspect of the data by aggregating all available features per entry with the 'logical or' operator. Since our features are binary, aggregation corresponds to summarizing the information into the set of events which occurred during the time period. In addition, we also considered an approach with a Bayesian Network (i.e., not a BNN). However, the BN model failed to converge due to the sparsity and high dimensionality of the data. Furthermore, we also trained many variants of XGBoost (Chen & Guestrin, 2016), but all of the boosted decision tree models produced random results. Therefore, we did not include the results for XGBoost below.

**Parameter settings/hardware.** All experiments were run on an NVIDIA P100. The code was implemented in PyTorch, using the Adam optimizer (Kingma & Ba, 2014) for all models, and trained for 400 epochs. The model with the lowest validation loss was selected for evaluation. The final hyperparameter settings are specified in Appendix A.

**Ablation study.** To understand the effect of the distribution on the BNN, we conduct an ablation study between the Gaussian BNN, Radial BNN, and their Spike and Slab versions. We provide results in Table 1. Clearly our proposed method provides better results in a number of metrics, including Specificity, Precision, F1, and FPR, which are important for Ransomware detection.

**Model Evaluation.** In Figure 1, we provide the ROC curves for the proposed model and several baselines. For our Radial Spike and Slab BNN method and the Gaussian BNN method, we display the distribution of each model's ROC curves, shaded in green, together with its mean value (e.g.,

| Statistics | Validation Set | | | | | | | Test Set: Future Time Period | | | | | | |
|---|---|---|---|---|---|---|---|---|---|---|---|---|---|---|
| | RNN-GRU | Neural ODE | FC | BNN: Gaussian | BNN: Radial | BNN: Gaussian Spike & Slab | BNN: Radial Spike & Slab | RNN-GRU | Neural ODE | FC | BNN: Gaussian | BNN: Radial | BNN: Gaussian Spike & Slab | BNN: Radial Spike & Slab |
| AUC | 0.85 | 0.83 | 0.83 | 0.83 | **0.88** | 0.87 | 0.87 | 0.70 | 0.73 | 0.77 | 0.75 | **0.81** | 0.79 | 0.77 |
| Specificity | 0.90 | 0.80 | 0.88 | 0.89 | 0.91 | 0.90 | **0.93** | 0.90 | 0.79 | 0.82 | 0.89 | 0.91 | 0.90 | **0.92** |
| Precision | 0.06 | 0.03 | 0.05 | 0.05 | 0.06 | 0.06 | **0.08** | 0.09 | 0.06 | 0.06 | 0.10 | 0.12 | 0.12 | **0.13** |
| FPR | 0.10 | 0.20 | 0.12 | 0.11 | 0.09 | 0.10 | **0.07** | 0.10 | 0.21 | 0.18 | 0.11 | 0.09 | 0.10 | **0.08** |
| FNR | 0.25 | 0.23 | 0.27 | 0.27 | 0.25 | **0.23** | 0.27 | 0.53 | **0.38** | 0.42 | 0.42 | 0.41 | 0.40 | 0.46 |
| FDR | 0.94 | 0.97 | 0.95 | 0.95 | 0.94 | 0.94 | **0.92** | 0.91 | 0.94 | 0.94 | 0.90 | 0.88 | 0.88 | **0.87** |
| Accuracy | 0.90 | 0.80 | 0.87 | 0.89 | 0.91 | 0.90 | **0.93** | 0.89 | 0.79 | 0.81 | 0.88 | 0.90 | 0.89 | **0.91** |
| Balanced Accuracy | 0.82 | 0.78 | 0.80 | 0.81 | **0.83** | 0.83 | 0.83 | 0.68 | 0.71 | 0.70 | 0.74 | 0.75 | **0.75** | 0.73 |
| $F_1$ | 0.11 | 0.06 | 0.08 | 0.10 | 0.12 | 0.11 | **0.14** | 0.15 | 0.11 | 0.12 | 0.18 | 0.20 | 0.19 | **0.21** |
| G-Mean | 0.82 | 0.78 | 0.80 | 0.81 | **0.83** | 0.83 | 0.83 | 0.65 | 0.70 | 0.69 | 0.72 | **0.73** | 0.73 | 0.70 |

Table 1: Ablation study for both validation set and the test set, which contains data from the future.

green line). Figure 1a shows that, with respect to the distribution of the ROC curves, our model outperforms the baselines on average, particularly in the region of small false positive rates, Figure 1b, which is the most important for ransomware detection. In addition, the Radial Spike and Slab BNN is able to provide a range of ROC curves which are significantly higher than the other baselines, if we consider the margins of the ROC distribution. Looking at the columns for the validation and testing set in Table 1, we see that proposed BNN model outperforms the baseline methods, w.r.t. to AUC, accuracy, G-Mean, and other statistics.

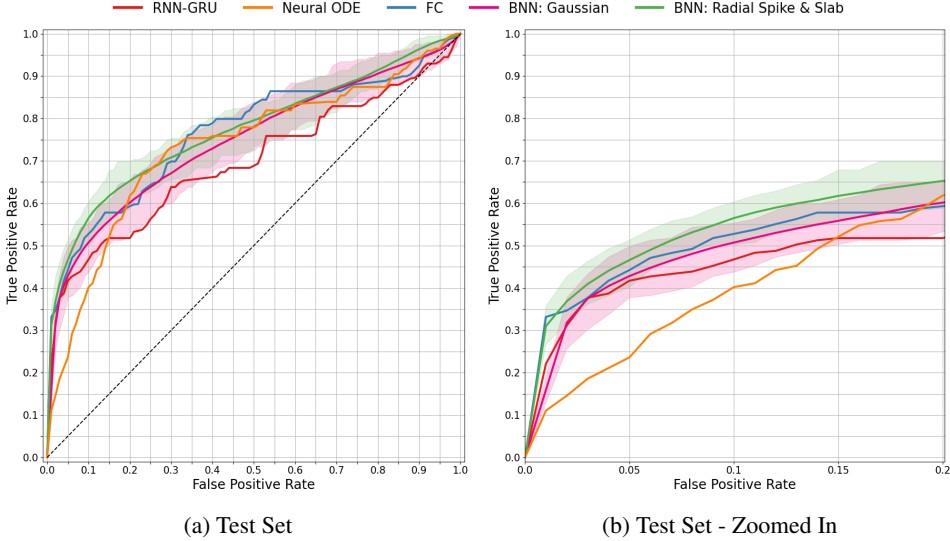

(a) Test Set              (b) Test Set - Zoomed In

Figure 1: We present the ROC curves for the new data from the future time period in the Test Set. Because the BNN is a probabilistic model, we show the distribution of the individual ROC curves (green shade) with the mean of this distribution (green line).

**Training and Test Times.** Training the Radial Spike and Slab Bayesian Neural Network in a single Azure-hosted Linux VM with an NVIDIA P100 for 400 epochs required 1 hour, 32 minutes and 53 seconds. The time required to evaluate the 15,340 samples in the test set was 19 seconds. However, to create a confidence interval (CI), the evaluation is repeated 100 times. Thus, 19 seconds corresponds to 100 evaluations. We re-ran inference on an NVIDIA A100 for all of the models to compare to 1 run of deterministic models. The results include 4.07 sec (1 iteration) and 10.37 sec (100 iterations for CI) for the BNN: Radial Spike & Slab (ours), 4.02 sec for the BNN: Gaussian, 7.72 sec for Fully Connected NN (FC), 3.12 sec for the Neural ODE, and 2.35 sec for the RNN-GPU.

**Feature Importance and Interpretation.**

We would like to understand which TTP features of the attack are considered to be important by our model when making a prediction whether an attack is ransomware or not. One way to do this is to investigate the posterior probabilities for the first layer weights of the BNN. However, while understanding the importance of the TTP features based on the BNN's trained weights conceptually makes sense, we instead follow a more well-known and established way to interpret the features of a general neural network, called Integrated Gradients (Sundararajan et al., 2017). Both methods are discussed in Appendix E. In Table 2, we present the subset of features which are the most important for our model to identify whether an attack is ransomware or some other type of attack based on Integrated Gradients. Sorting the values of Integrated Gradients, we find that the MITRE ATT&CK

features are significantly more important than the signature-based features. The "signature" in Table 2 is a low-level event generator from an analyst. As Table 2 shows, the MITRE events are much more important than the low-level signatures.

## 5 RELATED WORK

| Id | Feature representation |
|---|---|
| T1059.001 | Command and Scripting Interpreter, Powershell |
| T1105 | Ingress Tool Transfer |
| T1087 | Account Discovery |
| Signature | Suspicious activity was observed on this device |
| T1049 | System Network Connections Discovery |
| T1027.002 | Obfuscated Files or Information: Software Packing |
| T1566.001 | Phishing: Spearphishing Attachment |
| T1546.001 | Event Triggered Execution: Change Default File Association |
| T1218.003 | Signed Binary Proxy Execution: CMSTP |
| T1055.004 | Process Injection: Asynchronous Procedure Call |

Table 2: The Integrated Gradients method produces a score for each of the TTP features which indicates the importance of the feature for predicting whether the attack is ransomware (top) or another type (bottom). Features are ranked from the highest to lowest Integrated Gradients scores.

Recently, ransomware has become an active research area (Oz et al., 2022; McIntosh et al., 2021). Machine learning approaches have been proposed for the detection of ransomware attacks. A stacked, variational autoencoder is used to detect ransomware in the industrial IoT (IIoT) setting (Al-Hawawreh & Sitnikova, 2019). System API calls are used to detect ransomware using Decision Trees, a K-Nearest Neighbor classifier, and a Random Forest in (Sheen & Yadav, 2018). Takeuchi et al. (Takeuchi et al., 2018) also proposed using an SVM to detect ransomware using System API calls. Agrawal et al. (Agrawal et al., 2019) proposed a new attention mechanism on the input vector of an LSTM, an RNN and a GRU to improve the detection of ransomware attacks from API calls. An ensemble of network traffic classifiers are used to detect network packets and flows for the Locky family of ransomware in (Almashhadani et al., 2019). A Bayesian Network was the best performing flow-based classifier in this work while a Random Tree was the best for detecting packets in this work. HelDroid (Andronio et al., 2015) uses natural language processing techniques, along with static and dynamic analysis, to detect ransomware on mobile computing devices. Adamov and Carlsson (Adamov & Carlsson, 2020) use reinforcement learning to simulate ransomware attacks for testing ransomware detectors. Urooj et al. (Urooj et al., 2021) proposed an online classifier to predict early stage ransomware, but they do not provide any details for the classifier itself.

## 6 LIMITATIONS AND CONCLUSION

In this work, we propose the new Radial Spike and Slab Bayesian Neural Network and demonstrate that it outperforms the standard Bayesian Neural Network and other deep learning methods for the task of detecting ransomware attacks within the general class of all attacks, such as the dropping of commodity malware. The results can provide an early indicator of a potential ransomware attack for analysts to be able to confirm with additional investigation.

While the model is able to learn to distinguish between ransomware attacks and other attacks, the ROC curve indicates that it cannot be used by a fully automated system to completely disable computers or block network access due to a potential ransomware outbreak. However, since these attacks are being diagnosed by analysts, we believe that the model can alert these analysts about possible active ransomware attack on their network.

Given that ransomware attacks are relatively rare compared to the downloading of commodity malware, the amount of labeled data for these types of attacks is small. The size of our datasets from a production security service reflects this limitation. Fortunately, Bayesian computational methods such as Bayesian Neural Networks can be used for training and inference without overfitting in scenarios where the amount of labeled data is limited.

In addition to security problems, two significant areas, which focus on the analysis of high-dimensional sparse data, are Biostatistics and Genetics. While there are notable developments in sparse methods in those areas, our proposed method is novel for them as well. Once the source code is public, we hope that researchers from other fields will find the proposed model useful.

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

# A  HYPERPARAMETER SETTINGS

For reproducibility, this appendix provides the hyperparameter settings used for the proposed model as well as those for the baseline models. The best general hyperparameter settings from tuning are provided in Table 3. Other network hyperparameter are included in Table 4. The hyperparameters for the RNN, ODE, Fully Connected, and Bayesian models are provided in Tables, 5, 6, 7, and 8, respectively.

| Parameter | Value |
|---|---|
| train batch size | 100 |
| maximum num train epochs | 400 |
| hidden size | 706 |
| learning rate | 1e-4 |
| learning rate for probabilities | 1e-3 |
| Adam $\beta_1$ | 0.5 |
| Adam $\beta_2$ | 0.999 |

Table 3: General hyperparameters used for training the proposed Bayesian neural networks and baseline models.

| Parameter | Value |
|---|---|
| Include binary classification loss | True |
| Parameter for positive weight in the binary loss to represent imbalance of the data | 0.0068 |
| std used in likelihood term (or MSE) | 0.1 |
| dropout retention rate for discriminator | 0.9 |
| slope of leaky relu function | 0.2 |

Table 4: Network hyperparameters.

| Parameter | Value |
|---|---|
| Number of layers in ODE func in recognition ODE | 100 |
| Number of units per layer in ODE func | 0.0068 |
| ODE solver | Euler |
| ODE func units | 300 |
| ODE func rec num layers | 300 |

Table 5: ODE hyperparameters.

# B  RANSOMWARE ATTACKS

Ransomware attacks fall into two main categories, automated ransomware which include infamous cases such as WannaCry, and human operated ransomware (HumOR) conducted by actor groups such as REvil and a myriad of others. Although automated ransomware involves humans, the distribution of the payload usually does not involve human interaction. HumOR attacks, however, involve hands-on-keyboard activity, where an active human adversary has gained access to a network – whether through purchased access, malware, vulnerabilities, or other means – and progresses through the kill chain to escalate privileges, move laterally if possible, and distribute ransomware in the environment. Human operated attacks tend to be more severe, as the adversary is able to take steps to bypass protections and work to ensure the ransomware payload is executed successfully. Security solutions will actively monitor for these suspicious events across the different kill chain stages in a ransomware attack to detect and alert on the malicious behaviors.

Ransomware attackers will typically utilize multiple toolkits, custom malware, and scripts to conduct their activity more effectively. Often this can also entail multiple operators for different stages in the kill chain, such as with Ransomware-as-a-Service (RaaS) attacks. RaaS involves operators who work to create tools and provide access for vetted attackers – known as affiliates - to conduct the majority of the ransomware attack. Complicating matters, many of the tools ransomware attackers frequently use are open-source and have legitimate purposes, preventing outright detection and blocking unless the method of using the tools can specifically be classified as malicious.

| Parameter | Value |
|---|---|
| RNN cell type | GRU |
| GRU units | 300 |

Table 6: RNN hyperparameters.

| Parameter | Value |
|---|---|
| Number of hidden layers in FC network | 1 |
| Number of hidden units in FC network in FC network | 300 |

Table 7: Fully connected neural network hyperparameters.

There are several challenges for detecting and blocking ransomware attacks. First, there is a time criticality required for detection prior to the distribution and encryption of devices. Ideally, a good ransomware detection service can detect a ransomware attack prior to the encryption of any assets. This necessitates detecting the compromise as early in the kill chain as possible. However, the early stages of an attack do not necessarily have clear and specific implications of ransomware and can often mirror attacks that are not ransomware in nature. Second, although ransomware attacks are increasing and regularly reported in the news, they are still rare, and the labeled data is limited. Therefore, a ransomware detector must not overfit to sparse data. In addition, a ransomware detection service must have access to signals from a large number of computers or mobile devices in order to create datasets that can learn to detect important behaviors. Third, the system must generalize to handle polymorphism since the signals are polymorphic by their nature. Attackers may delay or reorder their activity, utilize open-source legitimate tools for malicious purposes, use polymorphic malware (e.g., backdoors) or scripts, or fast flux networks for command and control to avoid detection. Finally, the input signals are often weak and often do not indicate a ransomware attack on their own. An effective ransomware detection service must be able to combine these low-level signals in order to produce a successful high-level detection.

## C  THREAT MODEL

The BNN ransomware detector operates on data collected from the <the anonymized company's > currently operational backend security system. This system processes the low-level events which are generated by the device and stored in a cloud service, or alternatively on-premise, and, like all security services, this creates several areas which must be protected from attack. The low-level events are generated by the device itself in kernel mode of the operating system. The system assumes that the events are successfully generated, transmitted, and received by the cloud or on-premise backend service, and that the events have not been altered by the actor using a person-in-the-middle attack. Next, the system assumes that the events have not been altered once they have received and stored in the backend service itself. Thus, the system assumes that there are no successful data poisoning or insider threat attacks. Finally, the system assumes that the ransomware alerts are successfully transmitted to and correctly received by the customer's and the <the anonymized company's > analysis portals. It should be noted that all of these system components are operational today, and the proposed model only affects the processing component in the backend service.

## D  TEMPORAL PREPROCESSING

In this appendix, we investigate the sparsity of the raw dataset. In Figure 2, we show the distribution of the number of features that are set during each single one minute time step (i.e., time interval) in the raw data. In general, ten or fewer features are set during a single one minute time interval. A few one minute time steps have between 10 and 20 features that are set, while a small number of others have between 40 and 55 features set. Interestingly, we found that none of the time steps had between 20 and 40 features set.

| Parameter | Value |
|---|---|
| KL coefficient for VI | Graves |
| Method to compute KL | reparameterization |
| Number of samples to evaluate the test | 100 |

Table 8: Bayes hyperparameters.

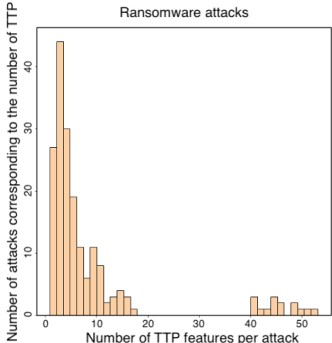

Figure 2: Count of the number of features available for 1, one minute time step of the ransomware data. The main mass contains less than 10 features per time point.

## E   FEATURE IMPORTANCE AND INTERPRETATION DETAILS

In this appendix, we discuss two methods to determine the most important features for the Radial Spike and Slab Bayesian Neural Network ransomware detection model. The first method we consider is to rank the posterior probabilities which are found in the first layer weights of the BNN. Recall that the core idea behind a BNN with Spike and Slab distributions is to learn a parameter $\theta_\pi$, which models the probability $S(\theta_\pi)$ of each node in the neural network to be included. Given that the first layer of the BNN is fully connected, we can consider $S(\theta_\pi)$ of the first layer as the importance of each TTP for our network. Since we suspect that not all TTPs are equally important, we expect to observe spikes in the learned $S(\theta_\pi)$. The results in Figure 3 confirm this hypothesis, where we start from the uninformative, uniform prior (left) and generate spikes in the learned $S(\theta_\pi)$ (right) after training. However, while understanding which TTP features are important based on the BNN's

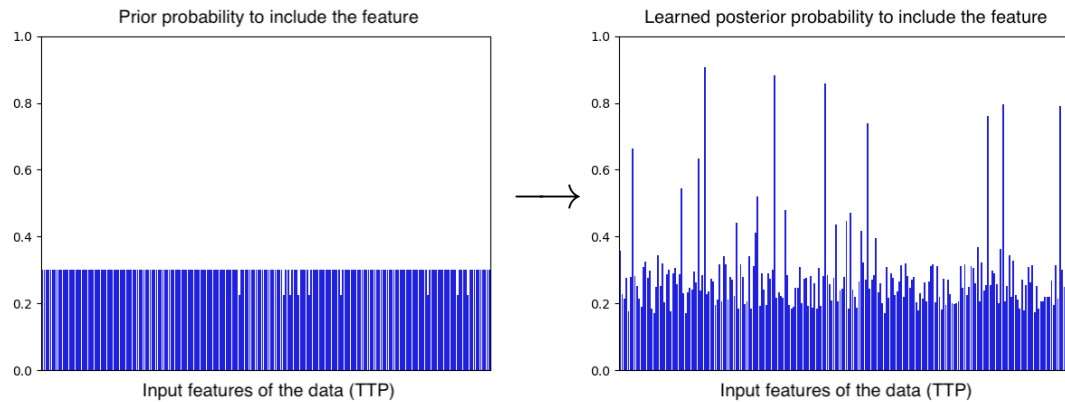

Figure 3: One of the approaches to evaluate the importance of the input features for a Spike and Slab BNN is to evaluate the learned posterior probability $S(\theta_\pi)$ (right). Note how different it is from the non-informative prior probabilities (left).

trained weights conceptually makes sense, we instead follow a more well-known and established way to interpret the features of a general neural network, called Integrated Gradients (Sundararajan et al., 2017) as the second method to rank the features.

Based on this procedure, we can generate an importance score for each feature given a trained network, and these scores are represented in Figure 4 for our model. In contrast to the Bayesian approach, this method also includes the signs of the feature scores. A positive attribution score means that a particular feature positively contributed to the final prediction of an attack being ransomware and a negative score indicates the feature was important for predicting non-ransomware attacks. The magnitude of the attribution score signifies the strength of the contribution. A feature which does not meaningfully contribute to the final output has a score of near zero.

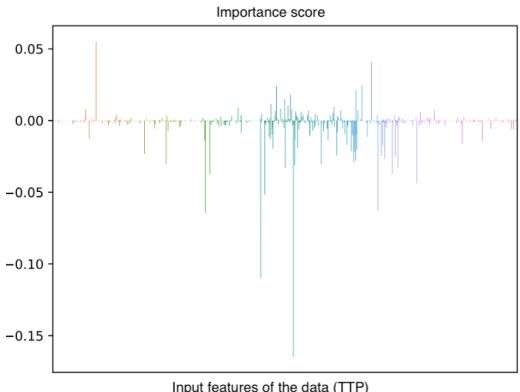

Figure 4: Applying the Integrated Gradients method generates scores which indicate the importance of the input features. A higher, positive score means that the feature is relevant for predicting data belonging to the ransomware class, while a negative score means the feature is more relevant to predicting the non-ransomware attack class.

## F    PROOF OF THEOREM 3.1

In this appendix, we provide the proof of Theorem 3.1.

*Proof.*

$$KL\left(q(w,\pi)\|p(w,\pi)\right)$$

$$= \int_\pi \int_w \log \frac{q(w,\pi)}{p(w,\pi)} q(w,\pi) dw d\pi$$

given that $q(w,\pi) = q(w|\pi)q(\pi)$ and $p(w,\pi) = p(w|\pi)p(\pi)$

$$= \int_\pi \left\{ \int_w \log \frac{q(w,\pi)}{p(w,\pi)} q(w|\pi) dw \right\} q(\pi) d\pi$$

given that $q(\pi) = Bern(\lambda_q)$ and $p(\pi) = Bern(\lambda_p)$

$$= q(\pi=0) \left\{ \int_w \log \frac{q(w|0)q(\pi=0)}{p(w|0)p(\pi=0)} q(w|0) dw \right\}$$

$$+ q(\pi=1) \left\{ \int_w \log \frac{q(w|1)q(\pi=1)}{p(w|1)p(\pi=1)} q(w|1) dw \right\}$$

$$= (1-\lambda_q) \left\{ \log \frac{1-\lambda_q}{1-\lambda_p} \int_w \delta_0(w) dw \right\}$$

$$+ \lambda_q \left\{ \log \frac{\lambda_q}{\lambda_p} + \int_w \log \frac{g_q(w)}{g_p(w)} g_q(w) dw \right\}$$

$$= (1-\lambda_q) \log \frac{1-\lambda_q}{1-\lambda_p} + \lambda_q \log \frac{\lambda_q}{\lambda_p}$$

$$+ \lambda_q \int_w \log \frac{g_q(w)}{g_p(w)} g_q(w) dw$$

$$= KL\left(Bern(\lambda_q)\|Bern(\lambda_p)\right) + \lambda_q KL\left(g_q\|g_p\right). \qquad \square$$

