# OpenReview forum: "Radial Spike and Slab Bayesian Neural Networks for Sparse Data in Ransomware Attacks"
_ICLR.cc/2023/Conference — Submitted to ICLR 2023_

### Official Review · Reviewer_zqwu · 2022-10-22

**Confidence:** 3
**Correctness:** 3
**Technical Novelty And Significance:** 3
**Empirical Novelty And Significance:** 2
**Recommendation:** 6

**Clarity, Quality, Novelty And Reproducibility:**

## Clarity
It is very clear.

## Quality
The theoretical justification looks right and the method is well motivated.

## Novelty
The method is moderately novel.

## Reproducibility
Code and datasets are not released so it is hard to reproduce.


**Strength And Weaknesses:**

## pros
-well-written and easy to read.

-the proposed method seems to be effective on ransomware attacks detection.

## cons
-only one internal dataset is used to evaluate the proposed method.
The proposed method might overfit that internal dataset to some extent.

-the paper might be better suited to a security conference.
Since the evaluation is only conducted on one dataset in ransomware attack, it is difficult to tell how generalizable the method is for other applications.

-appendix is not provided?
The paper refers to the appendix many times but the appendix is not provided? Or did I miss it?


**Summary Of The Paper:**

This paper proposes a method called “Radial Spike and Slab Bayesian Neural Network (BNN)” to detect ransomware attacks when the data is sparse and imbalanced. This method is a type of BNN with the posterior distribution represented by a mixture of distributions resulting in a Radial Spike and Slab distribution. The usage of this distribution is justified theoretically. For evaluation, the authors compare the proposed method against several baselines and variants on an internal ransomware dataset.

**Summary Of The Review:**

Good paper but evaluation is relatively limited to one internal dataset and it might not be best suited to ICLR.

---

> ### Author Response · Authors · 2022-11-16
> **Response**
>
> Q: only one internal dataset is used to evaluate the proposed method. The proposed method might overfit that internal dataset to some extent.
>
> A:  We have been looking for open datasets in other fields, like genomics, which are similar in features to our setup, namely highly sparse, binary and imbalanced data sets. Unfortunately, we did not find any since our expertise is in the security field. From the security perspective, it is hard to obtain real ransomware datasets that we can use to expand our experiment section since companies prefer to keep them private because of the security concerns. However, to evaluate the performance of our model and show that it does not overfit the data, we did test it on a completely new dataset collected in the future after the training data was collected. Thus, the test set included new methods of ransomware that were not included in the training data, and thus the model was not aware of these attacks. We showed that in this case our method outperformed the baseline methods.
>
> Q: the paper might be better suited to a security conference. Since the evaluation is only conducted on one dataset in ransomware attack, it is difficult to tell how generalizable the method is for other applications.
>
> A: Since we propose a new method with theoretical support, Bayesian Networks with Radial Spike and Slab distributions, we believe that ICLR is a better fit for our paper. While we understand the concern with limited datasets, we still hope that theoretical support of our paper with numerous SOTA baselines outweighs the limited datasets.
>
> Q: appendix is not provided? The paper refers to the appendix many times but the appendix is not provided? Or did I miss it?
>
> A: The appendix is in the main paper after references, page 13 in the submitted version.
>
> Dear reviewer, if we addressed your concerns, please, consider increasing the score.

---

### Official Review · Reviewer_qQmb · 2022-10-25

**Confidence:** 4
**Correctness:** 4
**Technical Novelty And Significance:** 2
**Empirical Novelty And Significance:** 2
**Recommendation:** 5

**Clarity, Quality, Novelty And Reproducibility:**

The paper's clarity is good, and the points are well articulated.

In terms of quality, the math is sound. However, more explanation is required for why slab and spike priors are appropriate for sparse signals since it's the main part of the approach. (instead of just citing the papers). The number of methods compared to each other is admirable.

However, the fact that there are just results on one dataset is a major negative point. I strongly recommend the authors find a dataset with a similar characteristic suitable for the proposed approach and run the comparisons again.

In terms of novelty, I acknowledge that the appropriate usage of spike and slab and the radial distribution choices are interesting. Also the theorem simplifies calculating the weight prior KLs. However, I'm not sure how these are important since the results are just reported on a single dataset. Such a method in my mind requires validation of every choice made throughout multiple datasets else it can look like a stack of clever heuristics.

**Strength And Weaknesses:**

The paper is well written, and although there is a lot of math the content is communicated well.

The proven theorem is interesting and simplifies calculating KL of weight priors to the approximate posterior distribution.

The relevant work is very properly cited making it easy to understand the paper.

The ROC curves show their method outperforms alternatives for ransomware attack detection. Their plot has confidence ranges that gives a better picture of their method's performance.

**Summary Of The Paper:**

The paper proposes a Bayesian Neural Network approach for ransomware attack detection.

The authors enumerate the challenges of ransomware detection: temporal, high-dimensional sparse signals with limited records and very imbalanced classes. They then propose using Bayesian Neural Networks is appropriate for these challenges and propose choices for the BNN setup suitable for ransomware attack detection.

An overview of their method is as follows:
1. They first convert each incident's low-level events to high-dimensional MITRE ATT&CK features.
2. To tackle signal sparsity, they make use of a commonly known mixture of priors with Spike and Slab components widely used in Bayesian Variable Selection.
3. To employ (2) in BNNs, they use a spike and slab prior to the weights. They prove a simple form for the KL divergence of two random variables with Spike and Slab distributions, later used in optimizing the BNN.
4. They reason Radial distribution is a better choice for the slab component.
5. To enable backpropagation through the sampling mechanism used in computing the loss, they use the parametrization trick for the slab radial component and substitute the Bernoulli part with Gumble-Softmax.

Then they compare their method with other approaches on one dataset. They also perform ablation studies and report runtime stats. They finally show how the BNN model can be used to give feature importance scores.

**Summary Of The Review:**

Overall I think the paper proposes a novel method addressing the challenges of the ransomware detection problem, but the experiments are not convincing enough that the proposals are impactful. It can be outperforming a regular BNN without these modifications on another dataset. Specifically, since the data is imbalanced and the number of positives in this single dataset is low.

---

> ### Author Response · Authors · 2022-11-16
> **Response**
>
> Q: However, more explanation is required for why slab and spike priors are appropriate for sparse signals since it's the main part of the approach. (instead of just citing the papers). The number of methods compared to each other is admirable.
>
> A: Thank you, we added the following clarifying text in the latest submitted version:
> “Namely, in our case, we would like to model sparse data with a sparse probabilistic Bayesian neural network.  Since only a portion of the input variables are relevant to the response variable, we want the weights to be represented as on/off switches to understand whether we should account for the input variables. Such a sparse Bayesian neural network can be represented by a `sparse` distribution on its weights, e.g., the mixture of priors with Spike and Slab components which have been widely used for Bayesian variable selection.”
>
> Q: However, the fact that there are just results on one dataset is a major negative point. I strongly recommend the authors find a dataset with a similar characteristic suitable for the proposed approach and run the comparisons again.
>
> A: We have been looking for open datasets in other fields, like genomics, which are similar in features to our setup, namely highly sparse, binary and imbalanced data sets. Unfortunately, we did not find any since our expertise is in the security field. From the security perspective, it is hard to obtain real ransomware datasets that we can use to expand our experiment section since companies prefer to keep them private because of the security concerns. However, to evaluate the performance of our model and show that it does not overfit the data, we did test it on a completely new dataset collected in the future after the training data was collected. Thus, the test set included new methods of ransomware that were not included in the training data, and thus the model was not aware of these attacks. We showed that in this case our method outperformed the baseline methods.
>
> Dear reviewer, if we addressed your concerns, please, consider increasing the score.

---

### Official Review · Reviewer_F9pu · 2022-11-02

**Confidence:** 4
**Correctness:** 3
**Technical Novelty And Significance:** 2
**Empirical Novelty And Significance:** 2
**Recommendation:** 3

**Clarity, Quality, Novelty And Reproducibility:**

The paper is fairly well written, and presents its key ideas in a lucid manner. The overall scope and originality of the paper is fairly limited however.

**Strength And Weaknesses:**

Strengths:
1) The setting of malware detection through the use of variational inference with BNNs is indeed an important topic of study, with possible immediate implications towards real-world security.
2) The proposed Radial Spike and Slab distribution for the BNN is indeed interesting, and helps simultaneously model a mixture distribution of discrete and atomless components to address the specific issues that are commonly seen to arise from the feature sets available in the setting of malware detection. The use of the Radial distribution (proposed by Farquhar et al., 2020) for the atomless component also helps circumvent the concentration of measure within an annulus as seen for multivariate gaussians in large dimensional settings.



Weaknesses:
1) The scope of the paper as presented is quite limited to the particular application of BNNs for detecting ransomware attacks amongst a set of more general malware attacks. This is quite disparate from that required in practical scenarios, wherein real-world security systems need to raise alerts amongst feature sets corresponding to (rare) occurrences of abnormal activity, amongst the general set of normal functioning parameters.
2) Furthermore, it is unclear why ransomware attacks alone are studied, given that other rare modes of malware attacks exist with corresponding sparse feature sets and comparable severity in ramifications arising from the security breach, all of which could potentially be addressed using BNNs as well. Thus, it is unclear why the detection of ransomware attacks alone amongst a set of malware attacks with equivalent or greater severity is required.
3)  The empirical evaluations presented are fairly limited, and could be significantly improved. For example, the paper states that variants of XGBoost produced random results. However, given that it is known that out of the sparse binary feature set of dimension 706, the 298 MITRE ATT&CK features are known to form the most crucial subset. Thus, for several models of lower complexity, this reduced feature set could be used to attain significantly improved baselines, by utilizing this domain-expert specific information to exclude near-irrelevant features.
4) Furthermore, the improvement in detection performance achieved using the proposed method as compared to baseline approaches such as BNN-Radial are extremely marginal, if any, as observed from Table-1. It is unclear if any improvement in detection performance can be established in a statistically significant manner, without performance statistics reported over multiple training instances of these networks with different random seeds.


Minor Typos:
The  abstract needs to be modified to follow the ICLR 2023 guidelines for typesetting and formatting.
The citation scheme used in the paper results in several instances where author names appear contiguously within sentences without demarcation.


**Summary Of The Paper:**

In this work, the authors propose to detect ransomware-based attacks amongst a general class of malware attacks using a Bayesian Neural Network (BNN). In such settings, the feature set available to subsequently perform inference or detection of vulnerabilities is quite sparse, and imbalanced. In order to address this, the paper proposes to utilize a new architecture for the BNN, namely using a Radial Spike and Slab BNN. Here, the weights are assumed to arise from a mixture distribution, with the first component being a dirac mass (or “spike”) and the second having an atomless support (or “slab”), namely a Radial distribution (Farquhar et al., 2020).

**Summary Of The Review:**

The paper studies the very narrow problem of the isolated detection of ransomware attacks amongst a larger class of malware attacks. Furthermore, the results obtained using the proposed Radial Spike and Slab BNN do not seem statistically significant in achieving improved detection over the baseline of Radial BNNs.

---

> ### Author Response · Authors · 2022-11-16
> **Response**
>
> Q: The scope of the paper as presented is quite limited to the particular application of BNNs for detecting ransomware attacks amongst a set of more general malware attacks. This is quite disparate from that required in practical scenarios, wherein real-world security systems need to raise alerts amongst feature sets corresponding to (rare) occurrences of abnormal activity, amongst the general set of normal functioning parameters.
>
> A: The anonymized event trace data is collected from a subset of computers in a global production system and represents the critical problem of determining if an attack is from some commodity malware, operating on one or more computers in the network, where an investigation may not be time sensitive, versus a human operated and/or automated ransomware outbreak across several or many computers in the organization’s network which has to be elevated and responded to immediately.
>
> The features that were provided are extremely helpful in detecting malicious activity (high precision for ransomware or commodity malware). Having a single feature activated for benign activity is very rare. As a result, computers which are acting normally very rarely produce false positive alerts (features). There are no known false positives from benign computers in our dataset. Therefore, this task is downstream from the normal security alert problem mentioned by the reviewer.
>
> Q: Furthermore, it is unclear why ransomware attacks alone are studied, given that other rare modes of malware attacks exist with corresponding sparse feature sets and comparable severity in ramifications arising from the security breach, all of which could potentially be addressed using BNNs as well. Thus, it is unclear why the detection of ransomware attacks alone amongst a set of malware attacks with equivalent or greater severity is required.
>
> A: The detection of ransomware attacks is one of, if not the, most important security challenges facing most organizations. As a result, this was the data that was collected and provided to us. There are rare modes of other types of malware, however, ransomware is in many ways a unique threat from malware attacks that presents its own specific challenges. Its ramifications from compromise often extend well beyond that of most other malware compromises. Though ransomware attacks can and frequently do emanate as a result of an initial malware compromise, its distribution and use is most typically human operated and markedly different than a malware attack. In human operated ransomware attacks, the adversaries can modify their behavior, tools, and payloads based on the environment, security products utilized, and other protections on the fly as needed. Evaluating ransomware attacks is a unique challenge that requires interpolating human threat activity and understanding how each technique used by an adversary may relate to others, and thus its own specific focus for a significantly more complex problem.
>
>
> Q: The empirical evaluations presented are fairly limited, and could be significantly improved. For example, the paper states that variants of XGBoost produced random results. However, given that it is known that out of the sparse binary feature set of dimension 706, the 298 MITRE ATT&CK features are known to form the most crucial subset. Thus, for several models of lower complexity, this reduced feature set could be used to attain significantly improved baselines, by utilizing this domain-expert specific information to exclude near-irrelevant features.
>
> A: We initially did a comprehensive initial study to exclude any irrelevant features. Some of the analyst-proposed event features are important (including one in top-10 features), and we would like to use all possible information in our setting to help combat the important problem of ransomware detection.
>
> Q: Furthermore, the improvement in detection performance achieved using the proposed method as compared to baseline approaches such as BNN-Radial are extremely marginal, if any, as observed from Table-1. It is unclear if any improvement in detection performance can be established in a statistically significant manner, without performance statistics reported over multiple training instances of these networks with different random seeds.
>
> A: The ROC curves are the most important performance metric of the proposed model, particularly at low false positive rates. We included the results in Table 1 for completeness, but these results are not as important as the ROC curves in Figure 1 since the corresponding threshold in Table 1 is set to 0.5 for the computation of these metrics. In other words, we would most likely not run the system with a threshold of 0.5.
>
>
> Dear reviewer, if we addressed your concerns, please, consider increasing the score.

---

### Decision · Program_Chairs · 2023-01-20

**Decision:**

Reject

**Justification For Why Not Higher Score:**

See my above descriptions of weaknesses.

**Justification For Why Not Lower Score:**

I recommend rejection.

**Metareview: Summary, Strengths And Weaknesses:**

In this paper, authors propose to detect ransomware-based attacks amongst a general class of malware attacks using a Bayesian Neural Network (BNN). Reviewers found the setting of malware detection via variational inference with BNNs an important topic of study. They also liked the paper organization and indicated that the paper is well written. However, several weaknesses were identified: perhaps the most critical one is that the empirical evaluations are fairly limited. The observed gains in some settings are also pretty marginal. Moreover,
the scope of the paper is quite limited (to the particular application of BNNs for detecting ransomware attacks amongst a set of more general malware attacks.) Given all, I think the paper needs a bit more work before being accepted.